# Diagnosis Assistance in Colposcopy by Segmenting Acetowhite Epithelium Using U-Net with Images before and after Acetic Acid Solution Application

**DOI:** 10.3390/diagnostics13091596

**Published:** 2023-04-29

**Authors:** Toshihiro Shinohara, Kosuke Murakami, Noriomi Matsumura

**Affiliations:** 1Department of Computational Systems Biology, Faculty of Biology-Oriented Science and Technology, Kindai University, Kinokawa 649-6493, Wakayama, Japan; 2Department of Obstetrics and Gynecology, Faculty of Medicine, Kindai University, Osakasayama 589-8511, Osaka, Japan

**Keywords:** colposcopy, acetowhite epithelium, diagnosis assistance, segmentation, deep learning

## Abstract

Colposcopy is an essential examination tool to identify cervical intraepithelial neoplasia (CIN), a precancerous lesion of the uterine cervix, and to sample its tissues for histological examination. In colposcopy, gynecologists visually identify the lesion highlighted by applying an acetic acid solution to the cervix using a magnifying glass. This paper proposes a deep learning method to aid the colposcopic diagnosis of CIN by segmenting lesions. In this method, to segment the lesion effectively, the colposcopic images taken before acetic acid solution application were input to the deep learning network, U-Net, for lesion segmentation with the images taken following acetic acid solution application. We conducted experiments using 30 actual colposcopic images of acetowhite epithelium, one of the representative types of CIN. As a result, it was confirmed that accuracy, precision, and F1 scores, which were 0.894, 0.837, and 0.834, respectively, were significantly better when images taken before and after acetic acid solution application were used than when only images taken after acetic acid solution application were used (0.882, 0.823, and 0.823, respectively). This result indicates that the image taken before acetic acid solution application is helpful for accurately segmenting the CIN in deep learning.

## 1. Introduction

Cervical cancer is caused by persistent infection with the human papillomavirus (HPV). It is the fourth leading cause of cancer death among women worldwide, affecting 604,000 people and killing 342,000 in 2020 [1]. There exists an effective vaccine for HPV. Vaccination rates are high in high-income countries and low in low-middle-income countries (LMICs). In Japan, vaccination rates are also low due to misconceptions about adverse reactions to vaccines [2]. Therefore, screening and examinations are crucial as prevention measures. Diagnosis of cervical cancer involves cytological diagnosis, colposcopy, and histological diagnosis. If cervical intraepithelial neoplasia (CIN), precancerous lesions of the cervix, is suspected in a cytological diagnosis, a colposcopy is performed by visually examining the cervix and applying an acetic acid or iodine solution with a magnifying glass. Applying acetic acid solution to the cervix changes the lesion’s appearance, making it white and appearing as a mosaic or dotted pattern. These changes enable us to estimate the presence and grading of CIN [3]. If CIN is suspected, a tissue sample is taken and a histological diagnosis is made. Because colposcopy is performed subjectively, the diagnostic result depends greatly on the competence of the gynecologist [4]. Thus, an accurate diagnosis is difficult for inexperienced gynecologists. In LMICs, where the number of experienced gynecologists is limited, accurate CIN classification not relying on the experienced gynecologists using machine learning methods would be very useful to screen cervical cancer and to point out the area on the cervix to sample for the histological examination.

Machine learning has already been used to help diagnose CIN [5], and various methods to classify CIN in colposcopic images have been proposed, such as methods using support vector machines [6,7] and deep learning [8,9]. Recently, huge datasets have been provided to compete for CIN grade classification performance [10] in kaggle, a well-known competition platform for data science and machine learning [11]. In addition, several studies have been reported on segmenting the lesions. For example, Kim et al. used the deep learning method SegNet to segment the lesions [12]. Yuan et al. also used a deep learning method, U-Net, to segment the lesions [13]. Yu et al. also segmented the lesions combining deep learning methods R-CNN, ASPP, and EfficientNet [14]. All the studies used only images taken after acetic acid solution application [15]. However, it is challenging for even gynecologists to identify the CIN from only the image taken after acetic acid solution application because the white appearance of the CIN resembles the normal squamous epithelium. They always consider the changes in the cervix appearance before and after acetic acid solution application in their diagnosis.

Therefore, in this study, we propose a deep learning method focusing on the changes before and after acetic acid solution application by segmenting the lesions in acetowhite epithelium, one of the representative types of CIN, in colposcopy. This study contributes to the establishment of an accurate diagnostic method for CIN.

## 2. Materials and Methods

### 2.1. Patients and Procedures for Colposcopy

Of the patients who underwent colposcopy for abnormal cervical cytology at the Kihankai Medical Corp. Flower Bell Clinic between June 2021 and March 2022, we selected those for whom the images taken before and after acetic acid solution application were available. For the colposcopy procedure in this study, first, the discharge was removed thoroughly, the cervix was observed with a colposcope (OLYMPUS Corp., Shinjuku, Tokyo, Japan), and the photograph was taken with a digital camera (E-PL5, OLYMPUS Corp., Shinjuku, Tokyo, Japan) connected to the scope with an exposure time between 1/100 and 1/40 s and an ISO sensitivity of 400. Next, a cotton ball soaked in 3% acetic acid solution was pressed against the cervix and left for 1 min. Then, the cotton ball was removed, the cervix was observed briefly (30 s) with a colposcope, and the photograph was taken again in the same manner. The focus was adjusted manually. The colposcopy was performed on all the patients by a gynecologic oncologist with more than ten years of experience.

### 2.2. Overview of the Lesion Segmentation Method Using Images Taken before and after the Acetic Acid Solution Application

The main feature of this lesion segmentation method is the use of images taken before acetic acid solution application in addition to images taken after acetic acid solution application for effective lesion segmentation. The changes in the appearance of the cervix can be recognized using images taken before and after acetic acid solution application. The workflow of the proposed method is shown in Figure 1. In order to recognize the changes at the same position in the images taken before and after acetic acid solution application, the image taken before acetic acid solution application is first aligned to the corresponding image taken after acetic acid solution application. Next, the image is scaled down, and the cervix is cropped manually. Note that this cropping is for performance evaluation in the experiment and is not essential for the proposed method. Both the obtained images taken before and after acetic acid solution application are input to U-Net as input data for training or testing. Finally, the lesion segmentation results are output. In the following sections, the details of the alignment and segmentation are described.

### 2.3. Alignment of Images Taken before the Acetic Acid Solution Application to Images Taken after the Acetic Acid Solution

In order to focus on the change in each pixel of the images taken before and after acetic acid solution application, the images taken before and after the application of acetic acid solution were aligned as a pre-processing step. We assumed that there was no significant change in the appearance of the cervix before and after the application of acetic acid solution due to the camera angle and the movement of the ostium of the uterus so that the alignment could be achieved by projective transformation. Note that the change in the appearance of the cervix here does not refer to the change in color or texture due to the application of acetic acid solution. A projective transformation is a transformation of an arbitrary rectangle into an arbitrary rectangle. To obtain the projective transformation matrix, four or more pairs of points at the same position (corresponding points) in images taken before and after acetic acid solution application are required. The appearance of the cervix in the images taken before and after acetic acid solution application changes significantly. Since this appearance change makes it difficult to find the corresponding points automatically by image processing, the corresponding points were determined accurately by visual inspection in this study. An example of the colposcopic images used in this study are shown in Figure 2. Figure 2a–c represent the image taken after acetic acid solution application, the image taken before acetic acid solution application, and the aligned image taken before acetic acid solution application deformed by the obtained projective transformation, respectively. In this method, the aligned images taken before acetic acid solution application are input to the U-Net with the images taken after acetic acid solution application. In Figure 2a,b, the numbered ‘+’ symbols are pairs of corresponding points set by the visual inspection.

### 2.4. Lesion Segmentation in Colposcopic Images Using Deep Learning with Colposcopic Images Taken before and after the Acetic Acid Solution Application

For semantic segmentation, a U-Net [16] is one of the most popular deep learning methods and is used for segmentation problems of medical images in various regions such as the brain, chest, and abdomen [17]. The U-Net has an encoder to extract the features of an object and a decoder to restore the original image size to obtain the segmentation results. In this study, we attempted to make the U-Net learn the changes in the appearance of the cervix before and after the application of acetic acid solution by inputting images both before and after the application of acetic acid solution to the U-Net. Labeling the lesions on the colposcopic images for the learning of the U-Net was performed based on the subjective judgment of the experienced gynecologist, and the labeled images were used as ground truth. Therefore, the ground truth did not necessarily correspond to the pathology results. In addition, since the purpose of this study was to confirm the validity of using the images taken before acetic acid solution application, only lesion segmentation was performed, and CIN grading classification was beyond the scope.

### 2.5. Experiments to Segment Acetowhite Epithelium

We experimented using actual colposcopic images to confirm the effectiveness of lesion segmentation using images taken before and after acetic acid solution application.

The specification of the computer used in the experiment is listed in Table 1. The resolution of the original colposcopic image was 4608 × 3456 pixels, and the image was scaled to 1152 × 864 pixels to reduce computational cost. Since the original image contained the speculum and the uterine wall, the input image was cropped to 480 × 640 pixels in size so that only the cervix was contained in the image in order to evaluate the performance of the lesion segmentation. In this study, the cervix was visually identified to ensure accurate cropping, and the image was manually cropped. Figure 3a, b, and c shows the cropped images taken before and after acetic acid solution application and their labeled image, respectively.

The conditions of the U-Net are listed in Table 2. The number of epochs, or the number of learning iterations using the dataset, was set to 500 epochs, at which the learning converges sufficiently. The purpose of the experiments was to confirm the effect of using images taken before and after acetic acid solution application compared to using only images taken after acetic acid solution application. Therefore, to keep the comparison as simple as possible, the hyperparameters, such as the number of epochs and mini-batch size, were fixed to be the same for both experiments. Since the number of cases was too small to prepare test images to evaluate the segmentation performance, the average value of the performance indices, accuracy, recall, precision, and F1 score from 450 to 500 epochs in the validation images was adopted for the performance evaluation. Note that the validation images do not affect the training because the hyperparameters were fixed and corresponded to the test images for evaluating the segmentation performance of the proposed method. Here, the accuracy is the rate of the correctly discriminated acetowhite epithelium and others in the entire image; the recall is the rate of correctly discriminated acetowhite epithelium out of the actual acetowhite epithelium; the precision is the rate of the actual acetowhite epithelium out of the discriminated acetowhite epithelium; and the F1 score is the harmonic mean of the recall and precision and is equivalent to the dice coefficient. Also, the segmentation performance at the epoch with the smallest value of the loss function was examined by the area under the curve (AUC) of the receiver operating characteristics based on the segmentation results to evaluate the potential performance of the proposed method. Since the segmentation of the acetowhite epithelium corresponds to a two-class classification, the acetowhite epithelium or not, the AUC was used as a measure of segmentation performance in this study. In this experiment, the encoder of the U-Net was VGG16 [18] pre-trained by ImageNet [19], a large-scale database of more than 20,000 categories for object recognition software research. The colposcopic images were transformed by randomly scaling them from 0.5 to 1.5 times, translating them vertically and horizontally by 10%, and flipping them horizontally in each epoch during training to accelerate learning. The transformed images were input to the U-Net, resulting in stochastic learning results. Therefore, 11 experiments of the lesion segmentation using the same dataset were conducted, and the results were averaged to evaluate the performance. The performance was validated by leave-one-out cross-validation due to the small number of cases. To confirm the effectiveness of the proposed method, we compared the results with those obtained by inputting only the images taken after acetic acid solution application to the U-Net.

## 3. Results

Forty cases were included in the study, of which ten were excluded. Of the ten cases, four cases were excluded because the cervix appearance remained the same before and after acetic acid solution application due to no lesions. When no lesion exists, it is impossible to evaluate the segmentation performance. Five of the cases were excluded because no corresponding point could be obtained. The reason corresponding points could not be obtained was that the appearance of the cervix changed significantly due to the movement of the ostium of the uterus or the camera angle. The other case was excluded because the shutter speed accidentally differed from the shooting conditions of this study. Consequently, we conducted the experiments using actual colposcopic images of 30 cases. Thus, the colposcopic images of 29 cases were used for training per case in the leave-one-out cross-validation. Details of the 30 patients included in the study are shown in Table 3.

Table 4 shows the results of inputting the images taken before and after acetic acid solution application to the U-Net (proposed method) and only the images taken after acetic acid solution application (control method). The entries in the table show the mean and standard deviation of 11 experiments of each performance index. The results of Welch’s t-test showed that the proposed method performed significantly better in all the evaluation indices except for the recall.

A box-and-whisker plot of each performance index averaged over the results of the 11 experiments per case is shown in Figure 4. The results of the proposed method and the control method were tested by Wilcoxon’s signed rank sum test. As a result, no significant differences were found, although the results of the proposed method were superior for all the indicators.

Table 5 shows the average AUC based on the segmentation results of the proposed and control methods at the epoch with the smallest loss function value in the validation data of 11 experiments. The results of Welch’s t-test showed that the AUC of the proposed method was significantly larger than that of the control method.

## 4. Discussion

To confirm the usefulness of using images taken both before and after acetic acid solution application, we compared the segmentation performance of the proposed method with that of the method using only the images taken after acetic acid solution application. In the results of the 11 experiments, the proposed method was significantly superior in all the evaluation indices except recall, as shown in Table 4. The acetowhite epithelium occurs on the area between columnar epithelium and squamous epithelium, called the squamous-columnar junction (SCJ), and becomes white by applying acetic acid solution. The normal columnar epithelium around the ostium of the uterus is reddish and barely changes its color when acetic acid solution is applied. The squamous epithelium surrounding the columnar epithelium is originally pale pink. However, when the color of the squamous epithelium is near-white, the color resembles the color of the acetowhite epithelium after the application of acetic acid solution. Therefore, it is difficult to distinguish the acetowhite epithelium from the squamous epithelium in only the image taken after applying acetic acid solution. Accordingly, it is challenging to determine from only the image taken after acetic acid solution application whether the white areas of the cervix in the image taken after the application of acetic acid solution were originally white due to squamous epithelium or temporally white due to acetowhite epithelium. Figure 5 shows an example of the acceptable segmentation results in the proposed mehtod. Figure 5a–d shows the manual segmentation result of the colposcopic image taken after acetic acid solution application (ground truth), the colposcopic image taken before acetic acid solution application, and the segmentation results of the proposed method and the control method, respectively. While the areas indicated by the white arrows in Figure 5d are incorrectly segmented in the control method, they are correctly segmented in the proposed method in Figure 5c. The advantage of using the images taken before and after cetic acid solution application is that it is possible to recognize the change in the cervix’s appearance by inputting the images taken before and after acetic acid solution application, as shown in Figure 5. Therefore, the proposed method is considered to have a significantly higher precision rate than that of the control method. A high precision rate means fewer normal sites are erroneously sampled, resulting in a lower physical burden on patients. On the other hand, although the recall rate of the proposed method was higher than that of the control method, there was no significant difference between the proposed and control methods. It seems that the recall rate tended to be high for both methods because the white areas of the cervix tend to be judged as acetowhite epithelium by the deep learning method, regardless of the use of the image taken before acetic acid solution application.

As shown in Figure 4, when the results were averaged over 11 experiments per case, there was no significant difference between the results of the proposed method and the control method. One of the reasons for these results is that the proposed method may be disadvantageous in some cases. For example, when the acetowhite epithelium is present in the SCJ area, whose color is initially close to white in the image taken before acetic acid solution application, the SCJ area is misjudged as squamous epithelium despite the presence of acetowhite epithelium. Figure 6 shows an example of the segmentation results. Figure 6a–d shows the manual segmentation result of the colposcopic image taken after acetic acid solution application (ground truth), the colposcopic image taken before acetic acid solution application, and the segmentation results of the proposed method and the control method, respectively. It can be found that the areas indicated by white arrows in Figure 6c are also relatively white in the image taken before acetic acid solution application. Thus, the areas were misjudged as normal squamous epithelium in the proposed method.

The AUC of the proposed method was significantly larger than that of the control method, as shown in Table 5. The results indicate that the proposed method learned features that better classify acetowhite and normal epithelium than the control method.

There are several limitations to this study. First, this is a retrospective single-center study using images taken by a gynecologist with a camera, and the sample size is small. Second, absolute segmentation performance cannot be discussed because of the small sample size. The U-Net encoder used in this study is VGG16, but other methods have excellent discrimination performance, such as ResNet [20] and EfficientNet [21]. There are also excellent semantic segmentation methods other than U-Net, such as DeepLabv3 [22]. It is also necessary to improve the robustness by tuning the hyperparameters of the deep learning method. To further improve the accuracy, specular reflection in the colposcopic image should be removed [23]. However, since this study aimed to confirm the usefulness of using images taken before and after acetic acid solution application, the proposed method was kept as simple as possible. The segmentation of lesions with higher performance will be the subject of future study. Third, due to the small sample size, only acetowhite epithelium was included. However, the proposed method is applicable to any CIN lesion, not just acetowhite epithelium. Fourth, since this study used a deep learning method to learn the gynecologist’s impression, labeled images based on the subjectivity of experienced gynecologists were used as the ground truth. Therefore, the results do not always agree with the results of the pathological examination. In addition, because of the small sample size, the CIN grading is beyond the scope of this paper. In future studies, we will consider learning pathological diagnosis, including CIN grade classification, by the proposed method.

## 5. Conclusions

In this paper, we proposed a method for segmenting lesions using deep learning to aid colposcopic diagnosis of CIN, focusing on the change in the appearance of the cervix before and after the application of acetic acid solution. In the proposed method, the aligned images taken before acetic acid solution application were input to the deep learning network, U-Net, for lesion segmentation in addition to the images taken after the application of acetic acid solution. In this study, 30 actual colposcopic images of acetowhite epithelium were used to compare the results of the proposed method with those using only images taken after acetic acid solution application. Although the originally whitish areas of lesions in the images taken before acetic acid solution application in some cases were misjudged as squamous epithelium in the proposed method, accuracy, precision, and F1 scores of the proposed method were significantly better than those produced when only images taken after acetic acid solution application were used. This result suggests that using images taken before and after acetic acid solution application can make the deep learning network learn the changes in the appearance of the cervix and that this method is, therefore, effective for accurate lesion segmentation.

## Figures and Tables

**Figure 1 diagnostics-13-01596-f001:**
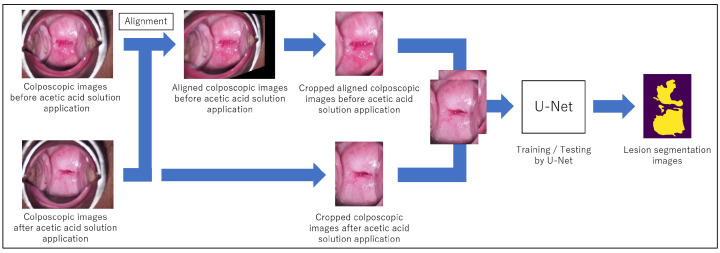
Workflow of the lesion segmentation method using images taken before and after acetic acid solution application.

**Figure 2 diagnostics-13-01596-f002:**
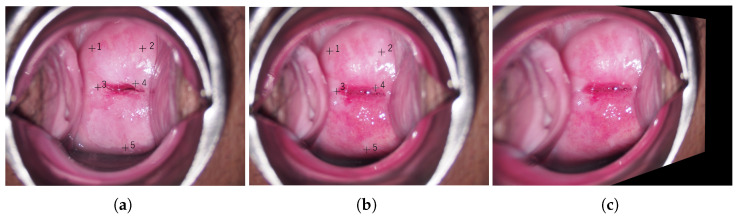
An example of colposcopic images; (**a**) Colposcopic image taken after acetic acid solution application; (**b**) Original colposcopic image taken before acetic acid solution application; (**c**) Aligned image taken before acetic acid solution application; numbered symbols are the corresponding points in (**a**,**b**).

**Figure 3 diagnostics-13-01596-f003:**
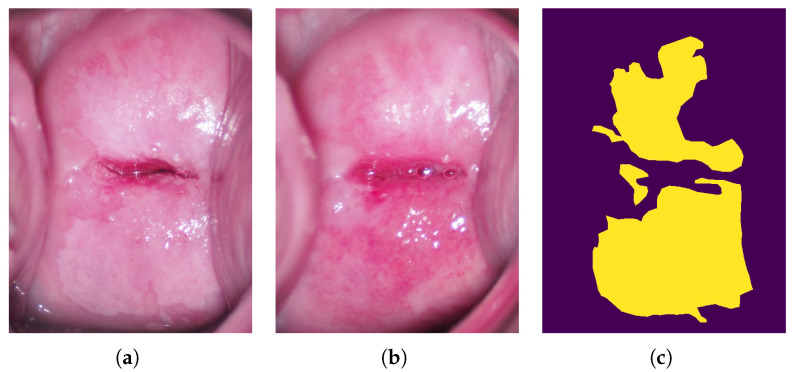
Cropped colposcopic images and their labeled image; (**a**) Colposcopic image taken after acetic acid solution application; (**b**) Colposcopic image taken before acetic acid solution application; (**c**) Their labeled image.

**Figure 4 diagnostics-13-01596-f004:**
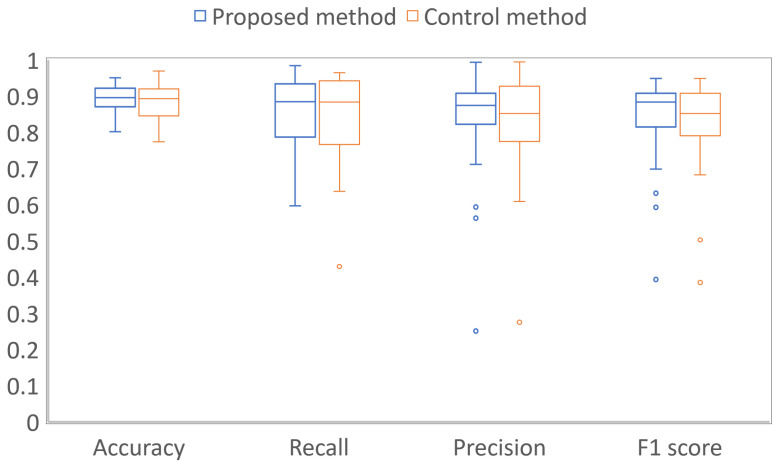
Results of averaging each performance index of the 11 experiments per case; left blue boxes indicate the results using the images taken before and after acetic acid solution application (proposed method), and right red boxes indicate results using only the images taken after acetic acid solution application (control method).

**Figure 5 diagnostics-13-01596-f005:**
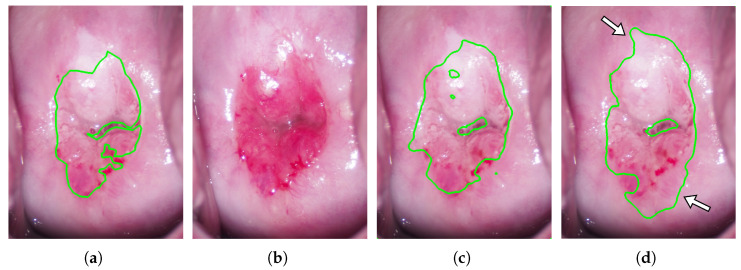
An example of acceptable segmentation results; (**a**) Manual segmentation result of the colposcopic image taken after acetic acid solution application (ground truth); (**b**) Colposcopic image taken before acetic acid solution application; (**c**,**d**) The segmentation results of the proposed method and the control method; the white arrows in (**d**) indicate areas that are incorrectly segmented in the control method.

**Figure 6 diagnostics-13-01596-f006:**
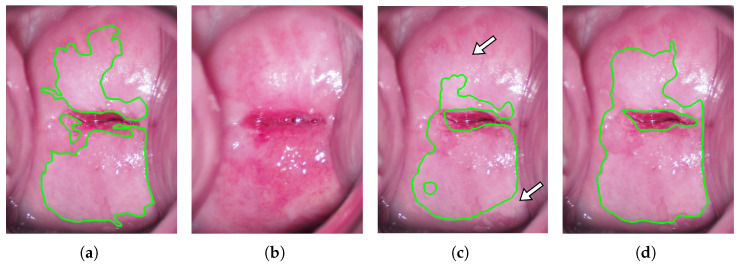
An example of poor segmentation results; (**a**) Manual segmentation result of the colposcopic image taken after acetic acid solution application (ground truth); (**b**) Colposcopic image taken before acetic acid solution application; (**c**,**d**) The segmentation results of the proposed method and the control method; the white arrows in (**c**) indicate areas that are incorrectly segmented in the proposed method.

**Table 1 diagnostics-13-01596-t001:** Specification of the computer used in this study.

Component	Model, Manufacturer
CPU	Core-i 9-10980XE, Intel Corp., Santa Clara, CA, USA
Memory	Crucial 32 GB × 4, Micron Technology, Inc., Boise, ID, USA
GPU	GeForce RTX 3090, NVIDIA Corp., Santa Clara, CA, USA

**Table 2 diagnostics-13-01596-t002:** Conditions of the U-Net deep learning method.

Mini-batch size	5
Number of epochs	500
Encoder	VGG16
Optimization method	Adam (Adaptive moment)
Loss function	Dice loss
Image augmentation	Scaling, translation, and horizontal flipping

**Table 3 diagnostics-13-01596-t003:** Patient characteristics.

		n	%
Total number		30	
Age (median)		33 (22–51)	
Cytology	ASC-US	14	46.7
	ASC-H	5	16.7
	LSIL	7	23.3
	HSIL	3	10.0
	AGC	1	3.3
High-risk HPV	positive	22	73.3
	not examined	8	26.7
Histology	no CIN	11	36.7
	CIN1	13	43.3
	CIN2	3	10.0
	CIN3	3	10.0
During pregnancy	yes	2	6.7
	no	28	93.3
Menopause	yes	1	3.3
	no	29	96.7

ASC-US: atypical squamous cells of undetermined significance, ASC-H: atypical squamous cells cannot exclude high-grade squamous intraepithelial lesion, LSIL: low-grade squamous intraepithelial lesion, HSIL: high-grade squamous intraepithelial lesion, AGC: atypical glandular cells, CIN: cervical intraepithelial neoplasia.

**Table 4 diagnostics-13-01596-t004:** Results (mean ± standard deviation) of using the images taken before and after acetic acid solution application (proposed method) and only the images taken after acetic acid solution application (control method).

Method	Accuracy	Recall	Precision	F1 Score
Proposed method	0.894 ± 0.004 **	0.855 ± 0.014	0.837 ± 0.013 *	0.834 ± 0.006 **
Control method	0.882 ± 0.004	0.842 ± 0.018	0.823 ± 0.009	0.823 ± 0.010

* p<0.05, ** p<0.01.

**Table 5 diagnostics-13-01596-t005:** The area under the curve (mean ± standard deviation) of the receiver operating characteristics based on the segmentation results of the proposed and control methods at the epoch with the smallest loss function value in the validation.

	Proposed Method	Control Method
AUC	0.943 ± 0.005 **	0.933 ± 0.008

AUC: area under the curve of the receiver operating characteristics; ** p<0.01.

## Data Availability

Not applicable.

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
