# Peer review of "Diagnosis Assistance in Colposcopy by Segmenting Acetowhite Epithelium Using U-Net with Images before and after Acetic Acid Solution Application"

_diagnostics, 2023, doi:10.3390/diagnostics13091596_

Round 1
Reviewer 1 Report
The use of the term ‘trials’ is misleading in this article since in clinical medicine a trial has a specific definition. The authors would be better to use the term ‘experiments’ or ‘observations’.
Line 91
This statement does not make sense. There are at least 3 grade of CIN.
Line 161
The colour of the normal squamous tissue of the cervical os is not white in colour, it is pale pink.
Line 163
The entire basis of colposcopy is the recognition of aceto-white change after the application of acetic acid. It is incorrect to state it is difficult to distinguish aceto white change after acetic acid from the squamous epithelium prior to application of acetic acid.
Line 211
A gynaecologist does not make a diagnosis of CIN. The gynaecologist has an impression that CIN is present. The diagnosis is made after a biopsy is taken to confirm the gynaecologists impression.
Author Response
The authors would like to thank you for your valuable comments.
Attached is our response to your comments.

Reviewer 2 Report
In this paper, "Diagnosis assistance in colposcopy by segmenting acetowhite epithelium using deep learning with images before and after acetic acid solution application" is presented. I have some major concerns about this paper:
1. The "deep learning" in the title is too large and not specific. It should be replaced by "Unet".
2. There are some syntax problems, such as "proposes" in the fourth line of the summary is replaced by "proposed".
3. In the latter part of the summary, it should be clear how much the proposed method has improved the quantitative indicators of the experiment?
4. The explanation of innovation points is not clear enough.
5. The workflow diagram of the proposed method should be added to the method.
6. Some sentences are too long, short sentences are recommended. For example, there is only one sentence in the 72-75th rows.
7.It should be stated, what is the original training sample in the experiment? How much has been amplified?
8. Some references in the article are missing, such as line 204.
9. Lack of conclusion.
10. There are only few two experimental comparison methods. It is recommended to add other machine learning comparison methods.
Author Response

(The authors gave the same response as above.)

Reviewer 3 Report
In this paper, authors propose a deep learning method to aid the colposcopic diagnosis of CIN by segmenting lesions using U-Net. One of the innovative aspects of their work is the simultaneous use of images before and after the application of acetic acid solution for segmentation. However, the experiments conducted on only 30 colposcopic images are not convincing, and the paper has several problems that need to be addressed.
1. To improve the robustness of the proposed method, I strongly recommend the authors use an independent test dataset for training, validation, and testing. Splitting the dataset into 7:2:1 for training, validation, and testing is a common practice in medical image processing.
2. Regarding the performance index, it is unclear if the accuracy mentioned refers to the dice similarity coefficient. If so, it would be better to correct the metric to dice.
3. Your task is to segment lesions of CIN. In general, AUC is an evaluation metric for classification tasks, and I am not sure that it is appropriate to use AUC here.
Author Response
Thank you for your careful review and helpful comments. Please see the attachment.

Round 2
Reviewer 2 Report
Comments and Suggestions for Authors. The authors have addressed most of my comments, which is much appreciated.
Author Response

(The authors gave the same response as above.)

Reviewer 3 Report
The manuscript has been revised well addressing the reviewer's comments.